# Oxygen-Sensitive Metalloprotein Structure Determination by Cryo-Electron Microscopy

**DOI:** 10.3390/biom12030441

**Published:** 2022-03-12

**Authors:** Mickaël V. Cherrier, Xavier Vernède, Daphna Fenel, Lydie Martin, Benoit Arragain, Emmanuelle Neumann, Juan C. Fontecilla-Camps, Guy Schoehn, Yvain Nicolet

**Affiliations:** 1Metalloproteins Unit, Univ. Grenoble Alpes, CEA, CNRS, IBS, F-38000 Grenoble, France; xavier.vernede@ibs.fr (X.V.); lydie.martin@ibs.fr (L.M.); juan.fontecilla@ibs.fr (J.C.F.-C.); 2Electron Microscopy and Methods Unit, Univ. Grenoble Alpes, CEA, CNRS, IBS, F-38000 Grenoble, France; daphna.fenel@ibs.fr (D.F.); arragain@embl.fr (B.A.); emmanuelle.neumann@ibs.fr (E.N.)

**Keywords:** metalloproteins, iron-sulfur cluster, cryo-electron microscopy, anaerobic environment

## Abstract

Metalloproteins are involved in key cell processes such as photosynthesis, respiration, and oxygen transport. However, the presence of transition metals (notably iron as a component of [Fe-S] clusters) often makes these proteins sensitive to oxygen-induced degradation. Consequently, their study usually requires strict anaerobic conditions. Although X-ray crystallography has been the method of choice for solving macromolecular structures for many years, recently electron microscopy has also become an increasingly powerful structure-solving technique. We have used our previous experience with cryo-crystallography to develop a method to prepare cryo-EM grids in an anaerobic chamber and have applied it to solve the structures of apoferritin and the 3 [Fe_4_S_4_]-containing pyruvate ferredoxin oxidoreductase (PFOR) at 2.40 Å and 2.90 Å resolution, respectively. The maps are of similar quality to the ones obtained under air, thereby validating our method as an improvement in the structural investigation of oxygen-sensitive metalloproteins by cryo-EM.

## 1. Introduction

Many d-block elements of the periodic table, also known as transition metals, are essential to all living organisms. They confer unique properties to proteins, thereby extending the repertoire of possible chemical reactions well beyond those made possible by amino acids alone. Metal-containing proteins can play key roles in many fundamental biological processes such as photosynthesis, respiration, or gas metabolism. A bioinformatic study comparing the genome of many organisms from the different kingdoms of life determined that approximately one third of all protein activities depends on transition metals [1]. Their importance in energy metabolism and the observed similarity between several organometallic cofactors and minerals led to the hypothesis that mineral surfaces played a key role in the emergence of life on Earth [2].

Iron is one of the major element on Earth’s crust and one of the most frequently found transition metals in metalloproteins [3]. Ferrous and ferric ions are often bound directly to specific amino acid residues or are bound by organic macrocycles, such as porphyrins, that tune their chemical properties. Iron ions can also combine with sulfide ions forming iron-sulfur ([Fe-S]) clusters. These clusters are probably the most versatile inorganic cofactors found in living organisms [4]. They exist predominantly as either rhombic [Fe_2_S_2_] or cubic [Fe_4_S_4_] clusters. These [Fe-S] clusters can fulfill different functions, including electron-transfer, redox and non-redox catalysis (such as Lewis acid-base chemistry), and environment sensing. They can also stabilize protein structures. More complex [Fe-S] clusters, sometimes combined with other d-block elements such as nickel or molybdenum are found in metalloprotein active sites where they catalyze very challenging chemical reactions. For example, H^+^, CO_2_ and N_2_ reduction are catalyzed by hydrogenase, carbon monoxide dehydrogenase (CODH) and nitrogenase, respectively [5]. These enzymes are key players in global carbon and nitrogen cycles and are a source of inspiration to chemists for the development of efficient and environment-friendly catalysts. [Fe-S] clusters are often sensitive to oxygen-induced degradation, leading to activity loss of the coordinating metalloprotein. Whereas some modifications can be limited to the metal center or its surrounding ligands, destruction of these metallocofactors in some cases leads to extensive structural rearrangements and quaternary structure modifications, notably through the dissociation of subunits (thereby increasing sample heterogeneity) [6,7]. For this reason, it has been necessary to develop standard methods and protocols operational in an anaerobic environment to carry out their structural and functional characterization. To this end, an increasing number of research groups has developed setups housed in anaerobic chambers in order to express, purify, analyze, and crystallize oxygen-sensitive proteins. These efforts have allowed for the description of a significant number of active site-bound metallocofactors and revealed their function [8,9,10]. Although X-ray crystallography is a very powerful technique in structural biology, it has several limitations. Besides the fact that crystallization is often a major bottleneck, crystal packing often limits the range of accessible protein motions, thereby precluding the in situ structural characterization of mechanisms that involve large conformational changes [11,12]. Crystallization is also challenging when working with multiprotein complexes that involve a large number of partners and/or when the interactions are only transient. In addition, keeping the sample stable during the crystallization experiment and getting the required milligrams of the protein to obtain useful crystals are also problematic. As an alternative, the recent and fast development of cryo-electron microscopy (cryo-EM) offers a way to solve structures at resolutions suitable for mechanistic studies without the above limitations. Indeed, the amount of sample needed is much smaller and no crystal is required. Furthermore, cryo-EM grids can be prepared in less than one hour after protein purification, compared to the days to weeks that may take for a crystal to appear. In addition, since each protein particle is observed, then individually selected and sorted in silico depending on its relative orientation as well as its conformation, or oligomeric state in the case of multimeric complexes, dynamic aspects could be addressed.

Over the last years, an increasing number of cryo-EM structures of metalloproteins and metalloprotein complexes have been solved [13,14,15,16]. Yet, to the best of our knowledge, the vast majority of them corresponds to samples prepared aerobically. In most cases, the samples were either oxygen-insensitive (or only slightly oxygen-sensitive) or oxygen sensitivity was not considered [17]. In the latter cases, aerobic cryo-EM grid preparation might have led to (unreported) metal center damage. Several X-ray crystallography groups expert in the study of oxygen-sensitive metalloproteins, are actively developing reliable methods to use negative staining electron microscopy and cryo-EM for protein structure solution [18,19].

Here we present a method to prepare grids based on a procedure we routinely use to harvest and freeze oxygen-sensitive metalloprotein crystals under anaerobic conditions [20,21]. This procedure is well-suited to a glovebox environment under conditions that have allowed the successful anaerobic characterization of oxygen-sensitive metal centers by X-ray crystallography. Here, we have applied it to determine the 3D structures of two model proteins by cryo-EM: apoferritin from horse spleen and pyruvate ferredoxin oxidoreductase (PFOR) from *Desulfovibrio africanus*. Apoferritin is highly stable, exhibits octahedral 432 symmetry and has a molecular mass of 444 kDa [22]. It corresponds to a gold-standard protein for cryo-EM and image analysis [23,24]. PFOR is a metalloprotein found in several anaerobic microorganisms where it is responsible for the reversible ferredoxin-dependent oxidative decarboxylation of pyruvate [25]. PFOR is a 267 kDa homodimer that contains three [Fe_4_S_4_] clusters per monomer (Figure 1). We chose PFOR since it is large enough to be easily studied by cryo-EM and it has been already studied by X-ray crystallography in one of our laboratories under the same anaerobic conditions [26,27,28]. The prepared cryo-EM grids for the proteins were of high quality, and we were able to obtain 2.40 Å and 2.90 Å resolution structures of apoferritin and PFOR, respectively, using a 200 kV Glacios^TM^ electron microscope. The latter model has a quality comparable with the available crystal structures [26,27,28] and displays intact [Fe_4_S_4_] clusters.

## 2. Materials and Methods

### 2.1. Protein Sample Preparation

Apoferritin from horse spleen was purchased from Sigma (reference number A3641). The protein was dialyzed against PBS, diluted four times from an 8.2 mg/mL solution, applied to grids and flash-cooled in the glovebox.

PFOR was purified as previously described [29]. The sample used in the present study was the same previously employed for our X-ray crystallography studies that was permanently stored in liquid nitrogen for the last twenty years [26,27,28]. The sample was thawed in a glovebox with less than 5 ppm ambient oxygen and injected to a Superdex 200 HR 10/30 size-exclusion column (GE Healthcare) in order to remove any protein aggregates. The chromatography experiment was performed at room temperature with 10 mM Tris buffer, pH 8.4 and the eluted protein was concentrated to 2.3 mg/mL.

### 2.2. Negative Staining

The PFOR sample was diluted to 0.3 mg/mL in 10 mM Tris buffer pH 8.4 and 3.5 μL of this solution were applied to the clean carbon side of a carbon/mica interface and subsequently stained using a 2% Sodium SilicoTungstate, pH 7.4 solution. Images were recorded using a Tecnai T12 Transmission Electron Microscope (FEI, Hillsboro, OR, USA) working at 120 kV with an Orius 1000 CCD camera under a low dose condition (less than 40 e^−^/Å^2^). Negative staining experiments were performed under the standard aerobic atmosphere just as a quality control test.

### 2.3. Cryo-EM Grid Preparation under Anaerobic Conditions

A Vitrobot Mark IV freezing system (FEI) was installed in a custom-made glovebox (JACOMEX, Dagneux, France) with an internal volume of around 915 dm^3^ and equipped with two front panel ovoid glove rings. (Figure 2a). Its right-side panel contains two airlocks, of 8 dm^3^ and 75 dm^3^ internal volume, which are connected to a primary vacuum pump and is equipped with an oxygen analyzer, a large-capacity gas purification system, a pressure regulator, and a refrigerating unit. N_2_ gas delivery to the glovebox is provided by evaporation of liquid nitrogen coming from an external reservoir. Under these conditions, the oxygen concentration inside this glovebox is routinely less than 1 ppm.

For the Cryo-EM grids preparation procedure, the Vitrobot Dewar is initially conditioned outside the glovebox. The following steps are carried out: (i) the brass cup is cooled with liquid nitrogen and subsequently asperged with gaseous propane that first liquefies and then solidifies; (ii) the Vitrobot nitrogen Styrofoam container is completely filled with freshly drawn liquid nitrogen in order to limit its contamination by oxygen from air (Figure 2b). The Vitrobot Dewar is subsequently introduced in the small airlock and set under vacuum for 3 min, which causes partial nitrogen solidification. Once inside the glovebox (Figure 2c,d), the small amounts of oxygen that could evaporate from the liquid nitrogen-filled pot are eliminated by a continuous flushing of the glove box atmosphere. To do so, an airstream is created, on the one hand, by suction of the vapors coming from the Dewar using the airlock pump and, on the other hand, by refilling of the nitrogen atmosphere through the gas delivery gate. The Vitrobot “Spider” tool is normally used to cool ethane. Here, it is used instead to heat and accelerate liquefaction of solid ethane by placing it a few seconds upside down on the brass cup. After both solid nitrogen and propane have liquified (Figure 2e,f), and the protein sample is introduced in the glovebox, the subsequent steps for flash-cooling the cryo-EM grids follow the same standard protocol used aerobically. The water used to maintain humidity in the grid-blotting chamber was extensively degassed through vacuum cycles and nitrogen gas bubbling to remove any trace of oxygen prior to be poured into the Vitrobot reservoir. After being placed inside cryo-boxes, grids were transferred back, with the Dewar, outside of the glovebox, through the airlock (without further vacuum cycle). The complete procedure must be completed in less than 30 min in order to keep the propane temperature below the vitreous/crystalline ice transition temperature of 136 K [30].

Cryo-EM grids (QUANTIFOIL^®^ grids Cu/Rh300 mesh, R 2/1) were glow-discharged (EMITECH, Montigny-le-Bretonneux, France) for 45 s at 30 mA under low vacuum before being introduced in the glovebox. Four μL of protein solution (about 2.0 mg/mL and 2.3 mg/mL for apoferritin and PFOR, respectively) were applied to the grid, which was then blotted and flash-cooled in liquid propane. The Vitrobot was set to operate at 22 °C and 100% humidity (blotting time 2 s; blot force 1). Finally, in order to recover the frozen grids, the Dewar was directly removed from the glovebox through the airlock.

### 2.4. Cryo-EM Data Acquisition and Image Processing

#### 2.4.1. Apoferritin

Apoferritin data were collected on a Glacios cryo electron microscope (Thermo Fisher Scientific, Waltham, MA, USA) operating at 200 KV, at a nominal magnification of 45,000× (corresponding to 0.899 Å/pixel at the specimen level), with a defocus range of −0.9 to −2.55 μm. Nanoprobe mode was used and the data collection was performed using parallel beam illumination. Movies were recorded using a K2 summit (Gatan, Pleasanton, CA, USA) direct electron detector operating in counting mode with the SerialEM software [31]. SerialEM was also used to perform objective astigmatism correction as well as coma-free correction. A series of 961 60-frame movies, were collected with a total dose of 60 electron ·Å^−2^ (1 electron·Å^−2^.frame^−1^) (Table 1).

Apoferritin reconstruction was performed using RELION 3.1.1 [32]. Each movie was first drift-corrected using MotionCor2 [33] (frames 2 to 60) and the contrast transfer function (CTF) was estimated using CTFFIND-4.1 [34]. Out of the 961 movies the 531 best ones (in term of resolution as determined by CTFFIND, and ice quality) were selected. Preliminary analysis was performed on 7 randomly selected micrographs. A total of 1690 particles were automatically selected using the Laplacian option of RELION and subjected to 2D classification. The 4 best classes were used as template to select automatically the particles from the 531 micrographs, giving a total of 133,493 particles. These particles were then subjected to 4 cycles of 2D classification before generating an initial 3D model with applied octahedral symmetry. The initial model was subsequently refined and an extra cycle of 3D classification (with octahedral symmetry imposed) enabled to select 66,655 particles that were used to calculate a new symmetrized and refined 3D map at 2.85 Å resolution. This refined map was then post-processed and polished using the default RELION options (refinement of both the asymmetrical and symmetrical aberrations and the per-particle defocus values were performed) with a B-factor value of −118.0 Å^2^, leading to the final 2.40 Å resolution map (0.143 gold standard FSC criteria; not shown).

#### 2.4.2. PFOR

PFOR data were collected on a Glacios cryo electron microscope (Thermo Fisher Scientific) operating at 200 KV, at a nominal magnification of 45,000× (corresponding to 0.899 Å/pixel at the specimen level), with a defocus range of −0.5 to −3.5 μm. The Nanoprobe mode was used and the data collection was performed using parallel beam illumination. Movies were recorded using a K2 summit (Gatan) direct electron detector operating in counting mode with the SerialEM software [31]. SerialEM was also used to perform objective astigmatism and coma-free corrections. A series of 1,304 60-frame movies, were collected with a total dose of 60 electron·Å^−2^ (Table 1).

PFOR reconstruction was performed using RELION 3.1.1 [32]. Each movie was first drift-corrected using MotionCor2 [33] (frame 2 to 60) using RELION and CTF was estimated using CTFFIND-4.1 [34]. We initially manually selected 1178 particles from 6 randomly selected micrographs to perform a preliminary 2D classification. The most promising 2D classes were used as templates to perform an autopicking on 50 micrographs. Next, 13,938 new particles were extracted and 2D sorted. The best classes were selected and used to calculate an initial 3D model. This model was subsequently used as template to perform an autopicking on the full dataset (913,217 particles). After a step of 3D classification, 180,868 particles corresponding to the best 3D classes were used to calculate a first 3D reconstruction. Post-processing of the corresponding map, using a B-factor value of −178.9 Å^2^, gave a resolution of 3.72 Å. Subsequently, refinement of both the asymmetrical and symmetrical aberrations and the per-particle defocus values was performed. Particle motion and the corresponding per-frame relative B-factors were estimated by Bayesian polishing before applying a second 3D refinement procedure. The map post processing using a B-factor value of −112.9 Å^2^, led to a resolution of 3.23 Å. A new round of aberrations and per-particle defocusing parameters refinement was performed, including estimation of the anisotropic magnification. After the last Bayesian polishing, 3D refinement was performed and angular distribution of the particles was verified (Appendix A). Post-processing, using a B-factor value of −91.7 Å^2^ gave a 2.90 Å resolution map using the FSC 0.143 cutoff criteria. C2 symmetry was applied in all of the 3D-refinement steps. Finally, the local resolution of the reconstruction was estimated using RELION 3.1.1 implemented procedure [35].

### 2.5. Model Building in the Cryo-EM Maps

Figures for apoferritin and PFOR structures were generated using Chimera [36] and PyMol [37].

#### 2.5.1. Apoferritin

A cryo-EM structure of horse spleen apoferritin (PDB code: 4V1W [38]; resolution: 4.70 Å) was used as a starting model. After initial manual building in COOT [39], the model was iteratively improved using PHENIX-real space refinement [40] and further manual building. Statistics validation was performed using both PHENIX validation tool and *MolProbity* until the statistics ceased to improve (Table 2). The map-to-model resolution was finally estimated at the 0.5 Fourier shell correlation (FSC) cutoff in PHENIX. The apoferritin cryo-EM map has been deposited to the EMDB (code: EMD-13487).

#### 2.5.2. PFOR

The crystal structure of the PFOR monomer (PDB code: 1B0P—chain A [26]; resolution: 2.31 Å) was initially fitted into the cryo-EM density as a rigid body. The dimer was then generated using the non-crystallographic C2 symmetry. Several iterative cycles of manual modification using COOT [39] and refinement with PHENIX [41] were necessary to build the complete model into the reconstructed cryo-EM density map (Table 2). Residues that did not match this map were removed from the model. Finally, those water molecules that were visible in the cryo-EM map and also present in the reference crystal structure (PDB code: 1B0P) were added to the model. The PFOR atomic coordinates and the corresponding cryo-EM map have been deposited to the PDB and EMDB, respectively (codes: PDB-7PLM and EMD-13493).

## 3. Results

### 3.1. Cryo-Electron Grids Preparation in Anaerobic Conditions

Placing a grid vitrification robot, such as the Vitrobot Mark IV freezing system (FEI/THERMOFISHER), in a glovebox is necessary to solve several technical issues before efficient cryo-EM grid preparation could be achieved. The main challenge came from the need to use of an airlock to enter samples in the glovebox. Indeed, the successive vacuum/flushing steps needed to remove all traces of oxygen would cause massive evaporation and partial loss of the cryogenic fluids contained in the cryo-EM grid flash-cooling Dewar (the central brass cup filled with liquid ethane or propane immersed in a pot with liquid nitrogen, see Figure 2). This procedure would negatively affect cryo-EM grid flash-cooling.

To overcome this problem, we took advantage of the protocol already developed in our laboratory to successfully flash-cool metalloprotein crystals under anaerobic conditions inside a glovebox [20]. This protocol involves the solidification of liquid ethane or propane in a vial placed in a liquid N_2_ bath before transferring it to the glovebox airlock -the melting temperature of ethane and propane being respectively 89.8 K and 85.4 K they can be solidified by liquid nitrogen (77.3 K). As solids, both ethane and propane can go through successive vacuum/flushing steps without undergoing significant evaporation/sublimation.

Under vacuum, the boiling point temperature of liquid N_2_ decreases, causing its partial evaporation (which is accompanied by heat loss). This, in turn, lowers the temperature of the remaining liquid nitrogen until it reaches its triple point at 63.1 K [42]. At this temperature, it no longer evaporates. Instead, it starts to solidify (Appendix A). This phase transition can be monitored by following the gas pressure inside the airlock while pulling a vacuum. Starting at atmospheric pressure, the airlock pressure decreases until it reaches a plateau at around 125 mbar. This represents the equilibrium between the vacuum generated by the pump and the evaporation of liquid nitrogen. When the nitrogen temperature reaches 63.1 K, evaporation stops and the pressure inside the airlock drops again to a final value of the primary vacuum. During the normal duration of this process, nitrogen does not fully solidify, and a fraction of it remains liquid (Figure 2c,d). Due to the significant nitrogen evaporation during the initial steps, its final volume is reduced by about one third. Still, there is enough nitrogen left to perform the cryo-EM grid flash-cooling under suitable conditions.

At this stage it is important to monitor the temperature of the flash-cooling system and the oxygen level in the glovebox to ensure a high-quality anaerobic cryo-EM grid preparation. In order to avoid formation of crystalline ice, ethane or propane must be kept at a temperature below 136 K, which corresponds to the vitreous/crystalline ice transition temperature of water [30,43]. We used a thermocouple wire to monitor the temperature over time and determined that, under our experimental conditions, the setup is functional for 30 min before requiring a refill. In practice, this allowed us to prepare four to six grids at a time (Figure 2f and Appendix A).

One major consideration when working with oxygen-sensitive samples is to avoid contamination by oxygen traces coming either from labware or solutions. Adsorbed oxygen can be partially removed by leaving the labware under vacuum in the airlock overnight, and oxygen dissolved in solutions can be removed through vacuum/flushing cycles and/or extensive bubbling with oxygen-free gas. We separately tested each element of the cryo-EM Dewar and determined that evaporation from the liquid nitrogen sample was the only significant source of oxygen contamination. Unfortunately, in our case none of the above-mentioned protocols can completely remove the liquid oxygen since liquid nitrogen has to be used rapidly. However, we reduced contamination by using liquid nitrogen immediately after getting it from the external reservoir -to limit oxygen contamination from contact with air- and by continuously flushing the glovebox with N_2_ gas. Furthermore, flushing was improved by (i) keeping the glovebox’s internal airlock door open during the experiment with the vacuum pump still running in order to create a moderate suction; and (ii) applying a continuous N_2_ gas flow to keep the pressure constant inside the glovebox. During the 30 min the liquid nitrogen remains cold enough to ensure optimal freezing of the cryo-EM grids, the oxygen level inside the glovebox goes from 1-5 ppm to less than 30 ppm. This final oxygen level remains in the range sometimes reached during anoxic crystallization/crystal harvesting in our laboratory with no observed damage. After recycling, the oxygen level in the glovebox was set back to 1–5 ppm.

Cryo-EM grids are usually prepared with liquid ethane [44]. However, our own experience of crystal flash-cooling for X-ray crystallography experiments, made us also consider propane for that procedure since it is easier to handle than ethane. The latter tends to foam when solidifying, but leads to the same flash-cooling results [20]. When testing both hydrocarbons, we found propane to be a better choice. Indeed, its melting temperature of 85.4 K is slightly lower than that of ethane (89.8 K), which should make it more efficient at avoiding ice crystal formation upon flash-cooling. More importantly, with propane a white haze at the surface of the Dewar was less often observed than when using ethane. This haze made grid recovery and transfer quite difficult. While we do not know the exact cause of this phenomenon, we assume that it might originate from traces of water vapor from the Vitrobot grid-blotting chamber.

### 3.2. Protein Reconstructions

#### 3.2.1. Apoferritin

Apoferritin is classically used to perform resolution tests in cryo-EM. Thanks to the high symmetry of this macromolecular assembly, the number of particles needed to calculate a 3D structure can be limited. Indeed, this sphere-like structure has 24 monomers related by octahedral symmetry. In addition, this sample is very robust and the protein structure quite rigid, as shown by the very high-resolution apoferritin models obtained by cryo-EM on state-of-the-art electron microscopes (1.2 Å in [45] and 1.25 Å in [23]. Consequently, it was a good control to check if our grid flash-cooling and handling procedures under anaerobic condition were correct and will give good results. This was indeed the case as we obtained a high-quality 2.40 Å resolution map of apoferritin. This resolution is similar to the one obtained using the classical freezing scheme on the same sample (not shown) on the same Glacios electron microscope equipped with a K2 summit electron detector. Better resolution could probably have been obtained using a state-of-the-art electron microscope, but that was not our aim. And, as seen in Figure 3 and in Table 1 and Table 2, both the fitting of the apoferritin structure to the EM map and the statistics are very good. This control clearly showed that with our setup and method, it is possible to prepare grids with ice layer thickness suitable to cryo-EM structure determination (Appendix A).

#### 3.2.2. PFOR

Cryo-EM grids of PFOR were prepared with the same sample previously used in our laboratory to obtain several X-ray structures of this enzyme [26,27,28]. Size-exclusion chromatography performed under anaerobic conditions revealed, as expected, the presence of homogenous protein dimers. Negative staining images obtained from an anaerobically prepared sample, but performed under aerobic conditions, confirmed that the PFOR solution was monodisperse and thus suitable for cryo-EM analysis (Appendix A). We subsequently prepared cryo-EM grids using our anaerobic setup (see Methods). The corresponding images were recorded using a Glacios (Thermo Fisher Scientific) electron microscope operated at 200 KV using a K2 summit direct electron detector (Gatan). Individual protein particles were clearly visible and well separated in most of the movies (Figure 4a), and the sample was then considered suitable for selection and 2D classification (Figure 4b). Several 2D classes display a clear C2 symmetry axis, as expected from protein dimers. As many as 180,868 particles were used to calculate the final 3D reconstruction at an estimated resolution of 2.90 Å by applying the gold standard FSC criterion (Figure 4c). Calculation of the local resolution indicates that the protein core, which represents about 83% of the volume of the reconstruction, is at a resolution between 2.75 Å and 3.05 Å, while the most exposed loops at the surface are at lower resolution ranging from 3.05 to 3.55 Å (Figure 4d).

An atomic model of the PFOR was subsequently built into the reconstructed 3D density map using the available crystal structure (PDB code: 1B0P [26]; 2.31 Å resolution) as a starting model. All of the PFOR structures published to date are very similar with a root-mean-square deviation (RMSD) of about 0.21 ± 0.05 Å. Similarly, the cryo-EM structure monomers display RMSDs of 0.53 and 0.54 Å, when compared to the crystal structure (PDB code: 1B0P). The most significant difference corresponds to a rotation of 1.6° between monomers when compared to the *D. africanus* PFOR crystal structures and up to 2.6° when compared to the enzyme from *Moorella thermoacetica* (Appendix A) [46].

The reconstructed cryo-EM density map was of good-enough quality to fit almost all of the protein residues (Figure 5a). As expected, less well-defined regions are found at protein surface loops. The largest disordered region corresponds to the first 14 residues of domain VII (residues 1170–1183), which consists of a domain-swapping extension (Appendix A). In addition, the Cys1195-Cys1212 disulfide bridge [47], the Ca^2+^ and Mg^2+^ ions and the thiamine pyrophosphate (TPP), essential for the protein activity, are clearly visible in the cryo-EM reconstruction (Appendix A). Their stereochemistry and coordination are similar to that of the corresponding crystal structure (Appendix A). More to the point, the three [Fe_4_S_4_] clusters are also perfectly defined in each monomer with high matching density (up to about 15 σ) (Figure 5b–d). The four iron and four sulfide atoms are observed in all clusters, and all of the metal ions are bound to their expected cysteine ligands. The stereochemistry is canonical and agrees well with the one observed in crystal structures (Appendix A). Only one of the cysteine ligands of the distal [Fe_4_S_4_] cluster displays an alternative conformation never seen in the published X-ray structures (Appendix A). All of the other cysteine ligands have the same stereochemistry as the previously published structures (Appendix A).

## 4. Discussion

Structural characterization of oxygen-sensitive metalloproteins requires specific equipment dedicated to sample preparation under anaerobic conditions. Several groups have developed setups and procedures to perform protein expression, purification, and crystallization inside gloveboxes [20,48]. Similarly, protocols have been developed to harvest and flash-cool crystals anaerobically for protein X-ray structural studies. The recent fast development of cryo-EM as a powerful technique to study single protein particles at resolutions comparable to those obtained by X-ray crystallography has emphasized the need to develop similar protocols to prepare cryo-EM grids anaerobically [18]. In the procedure described here, we adhered as much as possible to standard procedures performed outside a glovebox while implementing methods inspired from our anaerobic crystal harvesting protocol. In this way, we benefited from past developments while remaining open to future developments.

Working in a glovebox under a controlled atmosphere has the advantage of maintaining low levels of ambient humidity. Indeed, high humidity is problematic since the resulting frost may contaminate the grid during flash cooling. This, in turn, could limit the quality of the data to be collected. In our case, since the humidity level of the glovebox is kept at a very low level (around 1% relative humidity), we observe very little or practically no frost formation on the Dewar rim over the 30-min grid preparation procedure (Figure 2f). Working in a glovebox also makes it possible to lower the liquid nitrogen temperature to 63.1 K by carrying out several vacuum/flushing cycles in its airlock. [49]. This avoids nitrogen boiling inside the box and allows for excellent visibility during the whole process, facilitating grid handling after flash-cooling.

The grid preparation method reported here has been validated by the 2.40 Å resolution apoferritin cryo-EM structure, which is of similar quality to the ones generated using conventional aerobic conditions. The applicability of the method has been further confirmed by the determination of the 2.90 Å resolution PFOR cryo-EM structure. The cryo-EM refined model is very similar to previously published PFOR crystal structures, especially for several PFOR features ([Fe_4_S_4_] clusters, TPP, disulfide bridges, Ca^2+^). The only notable difference is an opening of the dimer interface in the latter, which is most likely due to crystal packing. The three [Fe_4_S_4_] clusters are clearly and fully visible in the cryo-EM reconstruction, and they are similar in shape and coordination to those observed in the crystal structure (Appendix A). This confirms that our method to prepare cryo-EM grids under anaerobic conditions, which preserves the integrity of the metal centers, is comparable to the one already developed for the crystallographic study of oxygen-sensitive proteins. Furthermore, the obtained structural details are similar to those acquired using standard cryo-EM grid preparation methods under aerobic conditions.

## 5. Conclusions

This study describes a procedure for preparing high-quality cryo-EM grids under anaerobic conditions to allow for the structural study of oxygen-sensitive biological samples by cryo-EM, leading to reconstructions of quality similar to those conventionally obtained in a normal atmosphere. Our method opens new paths for using cryo-EM for the structural characterization of single metalloproteins sensitive to oxidative damage (as well as of that of their complexes).

## Figures and Tables

**Figure 1 biomolecules-12-00441-f001:**
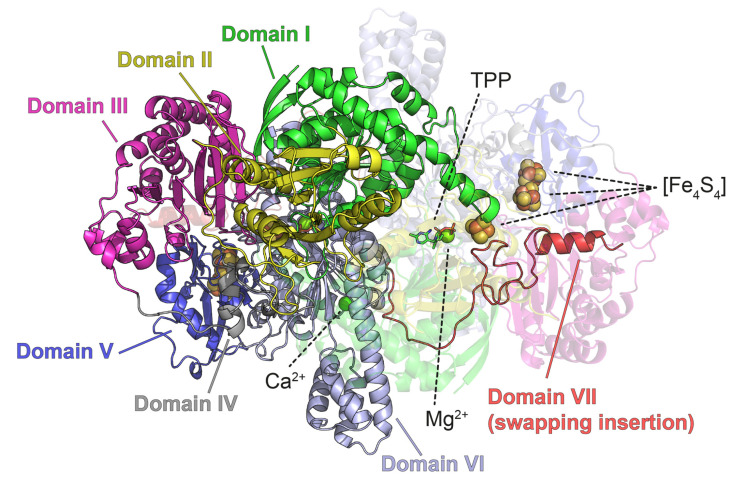
Crystal structure of PFOR (PDB code: 1B0P). The seven structural domains are colored differently as previously defined [26]: Domain I (residues 1–258), green; Domain II (residues 259–415), yellow; Domain III (residues 416–627), pink; Domain IV (residues 628–668), gray; Domain V (residues 669–785), blue; Domain VI (residues 786–1170), light blue; Domain VII (residues 1171–1232), red. TPP molecules are displayed in sticks; [Fe_4_S_4_], Mg^2+^ and Ca^2+^ are displayed with spheres. One of the monomers is displayed as a transparent image.

**Figure 2 biomolecules-12-00441-f002:**
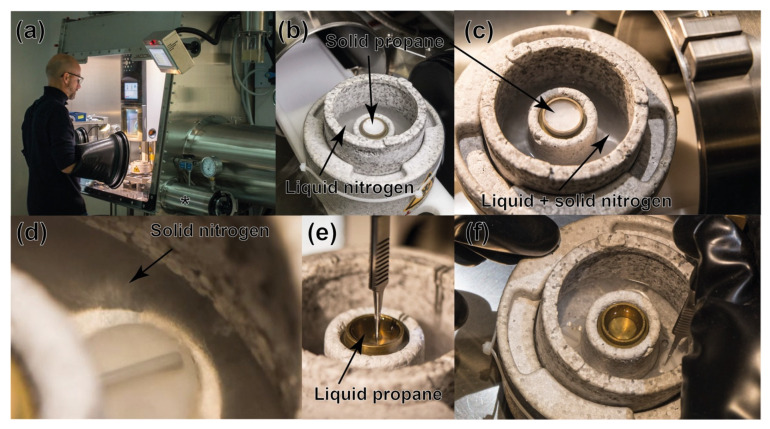
Cryo-EM grid preparation under anaerobic conditions. (**a**) View of the glovebox with the Vitrobot system inside. The symbol * shows the small airlock used to introduce the cryo-EM grids’ freezing Dewar. (**b**) View of the Dewar with its liquid nitrogen-filled pot and cup with solid propane, before its introduction to the airlock; and (**c**) treatment under vacuum inside the airlock causes the solidification of the nitrogen bath. (**d**) Close view of the pot with solid nitrogen (black arrow). (**e**,**f**) View of the brass cup with liquid propane where the cryo-EM grids will be immersed.

**Figure 3 biomolecules-12-00441-f003:**
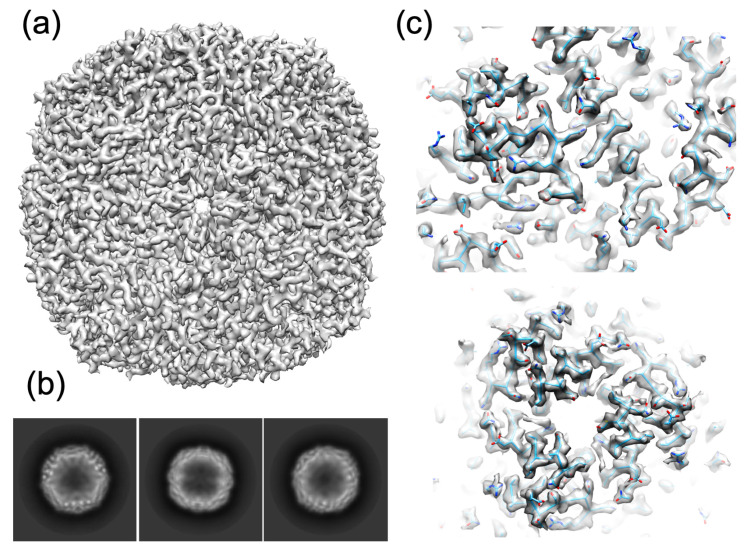
Final apoferritin 3D reconstruction contoured at 2.5 σ (**a**) Isosurface view of the apoferritin 3D reconstruction at 2.40 Å resolution. (**b**) Examples of 2D classes (respectively 8769, 7672 and 9251 particles are averaged in each class); (**c**) two views of the fit of horse spleen apoferritin (PDB code: 4V1W [38]) into the EM density map showing the good quality of the map.

**Figure 4 biomolecules-12-00441-f004:**
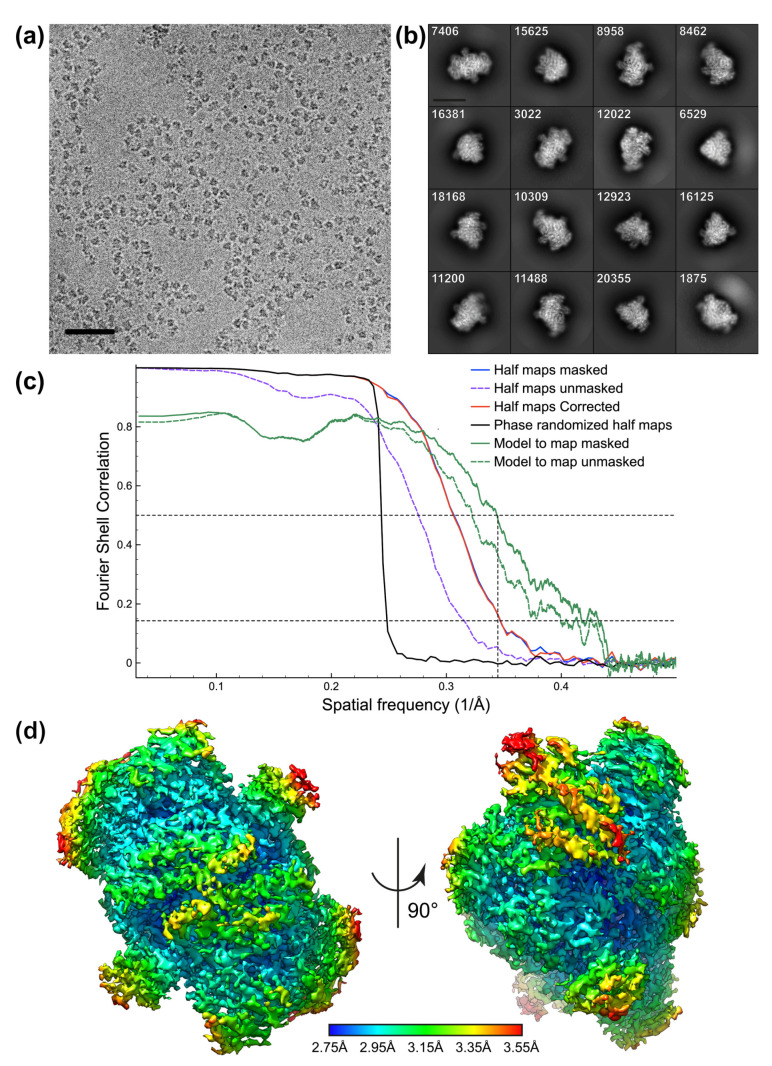
(**a**) Example of a PFOR collected micrograph (scale bar: 50 nm). (**b**) 2D classification of the selected PFOR particles (scale bar: 100 Å). Number of the particles in each 2D class are indicated. (**c**) PFOR Fourier-shell-correlation plots for independently refined half-maps (masked and unmasked, blue; corrected, red), phases randomized half maps (black) and full map versus model (Map to model, green). Horizontal black dashed lines indicate Fourier-shell-correlation values of 0.5 and 0.143. The vertical black dashed line indicates the 2.90 Å resolution. (**d**) Final PFOR reconstruction colored according to the local resolution: from dark blue (2.75 Å) to red (3.55 Å), contoured at 2.7 σ. A cross section of this reconstruction is presented in Appendix A.

**Figure 5 biomolecules-12-00441-f005:**
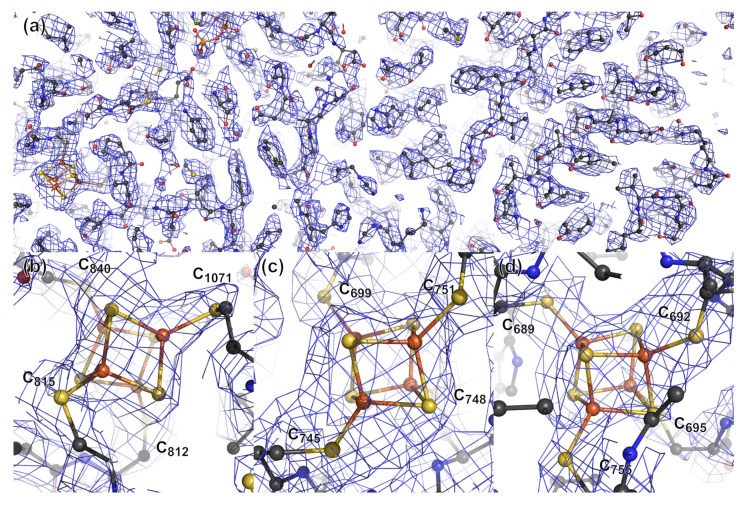
(**a**) View of the quality of the reconstructed cryo-EM density map and the fit of the protein model to it. Close view of the (**b**), proximal (**c**), median and (**d**) distal [Fe_4_S_4_] clusters of one PFOR monomer. The cryo-EM density mesh is colored blue and contoured at 5 σ.

**Table 1 biomolecules-12-00441-t001:** Cryo-EM data collection and processing.

	Apoferritin (EMD-13487)	PFOR (EMD-13493)
Magnification	45,000	45,000
Voltage (kV)	200	200
Total electron exposure (e^−^/Å^2^)	60	60
Defocus range (µm)	−0.9 to −2.55	−0.5 to −3.5
Pixel size (Å)	0.899	0.899
Symmetry imposed	Octahedral	C2
Number of movies	961	1304
Number of particles images in the final map	66,655	180,868
Map resolution (Å)	2.40	2.90
FSC threshold	0.143	0.143

**Table 2 biomolecules-12-00441-t002:** Cryo-EM refinement and validation statistics.

	Apoferritin	PFOR (PDB-7PLM)
Initial model used (PDB code)	4V1W	1B0P
Model resolution (Å)	2.40	2.31
Map sharpening B factor (Å^2^)	−118	−91.7
Model composition		
Chains	24	2
Protein residues	4 008	2 354
Water	321	114
Ligands	0	[Fe_4_S_4_] (6); Ca^2+^ (2); Mg^2+^ (2); TPP (2)
Average B factor (Å^2^)		
Protein	21.48	32.63
Ligand	/	20.48
Water	21.48	17.91
R.m.s. deviation		
Bond lengths (Å)	0.006	0.006
Bond angles (°)	0.952	0.691
Validation		
MolProbity score	1.22	2.50
Clash score	4.45	13.90
Rotamer outliers (%)	0.00	4.98
Ramachandran plot		
Favored (%)	99.34	95.47
Allowed (%)	0.66	4.53
Disallowed (%)	0.00	0.00
Model vs. Data		
CC (mask)	0.88	0.76
CC (main chain)	0.87	0.75
Mean CC for ligands	0.77	0.62

## Data Availability

The experimental apoferritin and PFOR cryo-EM maps can be downloaded from EMDB (codes: EMD-13487 and EMD-13493 respectively). The PFOR atomic coordinates can be downloaded from PDB (code: PDB-7PLM).

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
