# Peer review of "Oxygen-Sensitive Metalloprotein Structure Determination by Cryo-Electron Microscopy"

_biomolecules, 2022, doi:10.3390/biom12030441_

Round 1

Reviewer 1 Report

The manuscript by Cherrier et al. describes a technical advance enabling the cryo-EM characterization of oxygen-sensitive macromolecules under a non-oxidative environment, which they apply to study two macromolecules: (i) apoferritin as a standard sample for cryo-EM and (ii) PFOR as an oxygen-sensitive sample previously studied through X-ray crystallography. The method is of interest but the manuscript would benefit from improved presentation and discussion.

The starting hypothesis of the manuscript is that the function of oxygen-sensitive enzymes is altered unless kept under an oxygen-free environment. However, no comparison is made with the cryo-EM structure of PFOR obtained from grids prepared under standard conditions. As the time elapsed from sample application on the grid until vitrification is short, this may not harm the protein under study. Besides, their claim that ‘[Fe-S] clusters are often sensitive to oxygen-induced degradation, leading to activity loss of the coordinating metalloprotein’ (lines 47-49) should be supported with references.

In the introduction, the authors describe relevant properties that d-block metals confer to proteins, but omit that some play a key role in maintaining protein structure.

The authors claim that the main source of oxygen contamination is liquid nitrogen (line 278) but fail to describe how this was tested, also for other elements (line 277). Importantly, water used to humidify the grid-blotting chamber in the Vitrobot may also contain oxygen, which would be in direct contact with the sample just before it is vitrified. The authors should describe how they ensure absence of oxygen in the Vitrobot water reservoir. Besides, they should supply references to support their statement that 30 ppm oxygen (line 289) ‘does not affect metalloprotein active sites structures’ (line 290).

Regarding cryo-EM particle analysis, have the authors applied CTF-correction in the case of apoferritin, equivalent to what they did for PFOR? Also for apoferritin, have they applied octahedral symmetry in all steps, equivalent to C2 symmetry applied in the case of PFOR? Importantly, they authors should report the angular distribution of the PFOR map, as an additional quality control of the reconstruction.

The authors should improve description of their PFOR cryo-EM structure and its comparison to the reported crystal structure. First, a PFOR schematic domain representation should be included (for example in Fig 6) with indication of modelled regions, location of [Fe4S4] clusters, metals, TPP-binding residues, domain swapping insertion… This will help the general reader not familiar with PFOR structure to follow the text. Second, Fig S5 shows that both C1212 (forming a disulfide bridge) and C692 (binding a [Fe4S4] cluster) have been modelled with alternative conformations while the resolution is limited to 2.9 Å. The authors should reconsider their modelling of these two residues. Third, Fig S6 shows that the oxygen atom linking thiamine and pyrophosphate in TPP adopts a different configuration in the cryo-EM and crystal structures, but this is not mentioned in the text. Could this arise from the fact that resolution is limited to 2.9 Å? The authors should show a map around TPP and provide an expert opinion on this relevant difference. Fourth, Fig S5c shows a Ca2+ ion that is not mentioned in the text. Is metal coordination maintained with respect to that of the crystal structure?

Minor points:

  • Line 16. Please, spell out the PFOR acronym in the abstract for the general reader
  • Line 23 and 42. ‘d-block’ instead of ‘d block’
  • Line 66. ‘grids’ instead of ‘grid’
  • Line 103. Please, provide a brief description of the sample conditions during storage, which is more than 20 years
  • Line 178. ‘calculate’ instead of ‘calculated’
  • Lines 343-344. Resolutions are expressed with two significant digits after the decimal dot, while along the manuscript it is generally rounded to the first decimal place, which is more appropriate for reported resolutions
  • Line 416. ‘and’ instead of ‘et’
  • Tables 1 and 2. The symbol for thousands is not uniformly used
  • Table 2. What is SF4?

Author Response

Reviewer1

The manuscript by Cherrier et al. describes a technical advance enabling the cryo-EM characterization of oxygen-sensitive macromolecules under a non-oxidative environment, which they apply to study two macromolecules: (i) apoferritin as a standard sample for cryo-EM and (ii) PFOR as an oxygen-sensitive sample previously studied through X-ray crystallography. The method is of interest but the manuscript would benefit from improved presentation and discussion.

Answer: We thank the reviewer for his/her comments.

The starting hypothesis of the manuscript is that the function of oxygen-sensitive enzymes is altered unless kept under an oxygen-free environment. However, no comparison is made with the cryo-EM structure of PFOR obtained from grids prepared under standard conditions. As the time elapsed from sample application on the grid until vitrification is short, this may not harm the protein under study. Besides, their claim that ‘[Fe-S] clusters are often sensitive to oxygen-induced degradation, leading to activity loss of the coordinating metalloprotein’ (lines 47-49) should be supported with references.

Answer: The main goal of our manuscript was to report a reliable method to prepare cryo-EM grids of high quality under anaerobic environment to preserve metalloprotein samples from possible oxygen-induced damages. We did not pretend to study such damages through a comparative structural analysis. It is already well documented and broadly admitted in the metalloprotein community that oxygen alters transition metal centers through oxidative damages, often leading to reactive oxygen species (ROS) production. We have now added references as requested by the reviewer to support this point. Furthermore, it is also established that metallocofactor integrity can be preserved provided experiments are performed in dedicated anaerobic chambers such as the ones we have developed in our group. This has notably been proven by the different crystal structures we solved over the last twenty years. Yet, harvesting such crystals usually requires to prevent oxygen exposure. In our experience, if such crystals are taken out of the glovebox and quickly flash frozen in air, we often observe an increase of the B-factor values of the metallocentre and surrounding residues, leading to a blurring in the structural information provided by such structures. We can reasonably assume that the same would occur when the 1-3 mL drop is deposited to a cryo-EM grid, even if the time-course of the preparation is short. Nevertheless, we do not expect a collapse of the overall structure. Yet, in some cases, the important structural information is located around the metallocenter, notably when such center is located at the interface of different subunits. In those cases, the structural integrity of the metallocenter is necessary to preserve the quaternary structure. Therefore, it is important for the metalloprotein field, to have reliable protocols to prepare high quality cryo-EM grids in oxygen-free environment to tackle structural investigation of such systems.

In the introduction, the authors describe relevant properties that d-block metals confer to proteins, but omit that some play a key role in maintaining protein structure.

Answer: We are now mentioning the role of iron-sulfur clusters in stabilizing protein structure.

The authors claim that the main source of oxygen contamination is liquid nitrogen (line 278) but fail to describe how this was tested, also for other elements (line 277). Importantly, water used to humidify the grid-blotting chamber in the Vitrobot may also contain oxygen, which would be in direct contact with the sample just before it is vitrified. The authors should describe how they ensure absence of oxygen in the Vitrobot water reservoir. Besides, they should supply references to support their statement that 30 ppm oxygen (line 289) ‘does not affect metalloprotein active sites structures’ (line 290).

Answer: Every component of the setup was tested separately. We initially suspected the polystyrene bowl itself but it turned out that oxygen contamination only happened when “aged” liquid nitrogen was introduced in the glovebox. In fact, the liquifying temperature for oxygen is 90K, while that of nitrogen is 77K. Hence oxygen liquifies with time into liquid nitrogen and can therefore be slowly released upon evaporation. The water used in the grid-blotting chamber was degassed prior to be poured into the reservoir. Therefore, we can exclude any oxygen contamination from it. We corrected our statement regarding absence of oxygen damage below 30ppm oxygen. Our own experience when purifying and crystallizing metalloproteins indicated that below that value, no oxygen-induced damage was observed (monitored by different spectroscopic methods such as Electron Paramagnetic Resonance or UV-visible). We do not know what is the limit and this is most probably sample-dependent, but at least, stopping at 30 ppm prevents significant damages.

Regarding cryo-EM particle analysis, have the authors applied CTF-correction in the case of apoferritin, equivalent to what they did for PFOR? Also for apoferritin, have they applied octahedral symmetry in all steps, equivalent to C2 symmetry applied in the case of PFOR? Importantly, they authors should report the angular distribution of the PFOR map, as an additional quality control of the reconstruction.

Answer: We have changed the text accordingly to better explain the procedure, notably that we indeed applied the symmetry all along the process for apoferritin. A figure with the angular distribution of the PFOR reconstruction has been added in the supplementary material (Figure S1).

The authors should improve description of their PFOR cryo-EM structure and its comparison to the reported crystal structure. First, a PFOR schematic domain representation should be included (for example in Fig 6) with indication of modelled regions, location of [Fe4S4] clusters, metals, TPP-binding residues, domain swapping insertion… This will help the general reader not familiar with PFOR structure to follow the text.

Answer: This has been done. A new figure has been added (Figure 1).

Second, Fig S5 shows that both C1212 (forming a disulfide bridge) and C692 (binding a [Fe4S4] cluster) have been modelled with alternative conformations while the resolution is limited to 2.9 Å. The authors should reconsider their modelling of these two residues. Third, Fig S6 shows that the oxygen atom linking thiamine and pyrophosphate in TPP adopts a different configuration in the cryo-EM and crystal structures, but this is not mentioned in the text. Could this arise from the fact that resolution is limited to 2.9 Å? The authors should show a map around TPP and provide an expert opinion on this relevant difference. Fourth, Fig S5c shows a Ca2+ ion that is not mentioned in the text. Is metal coordination maintained with respect to that of the crystal structure?

Answer: Following Reviewer 4’s request, we further refined the PFOR model. We notably reconsidered the refinement parameters leading to better statistics and slight structural modification such as the TPP geometry, which now corresponds to that of the previously reported crystal structures. The corresponding map around TPP is displayed in figure S8a. Regarding the alternative conformations of C1212 involved in the disulfide bridge, the density was not clear enough to confirmed its existence and we removed it as was suggested by the reviewer. Conversely for the C692 residue, the clear density around residue C692 is still observed, supporting an alternative conformation of the cysteine. We therefore decided to conserve it. The Ca2+ ion is now mentioned in the text.

Minor points:

  • Line 16. Please, spell out the PFOR acronym in the abstract for the general reader

Answer: Pyruvate-ferredoxin oxidoreductase has now been indicated

  • Line 23 and 42. ‘d-block’ instead of ‘d block’

Answer: Done

  • Line 66. ‘grids’ instead of ‘grid’

Answer: Done

  • Line 103. Please, provide a brief description of the sample conditions during storage, which is more than 20 years

Answer: Done

  • Line 178. ‘calculate’ instead of ‘calculated’

Answer: Done

  • Lines 343-344. Resolutions are expressed with two significant digits after the decimal dot, while along the manuscript it is generally rounded to the first decimal place, which is more appropriate for reported resolutions.

Answer: Resolutions values had been changed to two digits formats.

  • Line 416. ‘and’ instead of ‘et’

Answer: Done

  • Tables 1 and 2. The symbol for thousands is not uniformly used

Answer: Done

  • Table 2. What is SF4?

Answer: SF4 is the standard 3-letter code for [Fe4S4] cluster as defined in the Protein Data Bank. It has now been replaced by [Fe4S4] in Table 2.

Reviewer 2 Report

In this study, Cherrier et al. describe their setup for cryo-EM grids preparation in an anaerobic chamber ("glove box") for the purpose of studying the structure of oxygen-sensitive metalloproteins.

The paper reads well and introduces nicely and clearly for a non-expert audience the importance and specificity of the biochemistry of the metalloproteins and the methodological challenges encountered when studying them, particularly the necessity of working in a anaerobic environment.

The description of the setup for cryo-EM grid freezing in an anaerobic chamber (adapted from a cristallisation setup also developed by the lab) is clear and easy to follow for all people with minimal knowledge of the cryo-EM workflow. The important steps, specific bottlenecks and the ways to overcome them are well explained and described.

As a proof of concept, the authors also prepared and solved the structure of a well-known test sample aka apoferritin (the "lysozyme" of cryo-EM) to show the ability of the installation to produce "good" grids; and they also prepared and solved the structure of an oxygene-sensitive metalloprotein (PFOR) to prove that the anaerobic environment is working. Both cryo-EM structures were at rather high resolution (below 3 Å) and the procedures used and described (grid preparation, data collection, single particle analysis and model building) correspond to a "state-of-the-art" workflow.

Major points:

  1. As a non-expert of the metalloproteins field, I think it is important to describe what kind of damage(s) could be expected from the oxygen action on a metal center; will it be destroyed? modified? What should we expect to see (or not see) in a (cryo-EM) structure done in aerobic conditions? What would be the minimal resolution to achieve to clearly see and assess such damages?
  2. On a similar line; how well is PFOR suitable for such a study? It is mentioned that it is oxygen-sensitive but to what extend? I understand that a cristallisation experiment can last several days, even weeks, but freezing cryo-EM grids takes ~30 minutes: is this time frame still relevant in term of oxygen exposure, i.e. is that enough to "damage" the metal centres of the PFOR dimer?
  3. In section "3.2.1 Apoferritin". I strongly disagree with the statement saying that the resolution is limited by the EM setup and that better resolution could be achieved by using a "state-of-the-art" electron microscope. One needs to compare what is comparable: the very-high resolution (sub-1.5 Å resolution) apoferritin structures cited by the authors are from a complete different source and definitely not horse spleen protein purchased from SIGMA. A quick research in the EMDB (filtering the horse apoferritin structures) shows on the opposite that the apoferritin structure reported in the present paper is on the high resolution side for such a sample, none of them going below 2.1-2.2 Å! This paragraph needs to be modified.

Minor points:

  1. line 80: "prepared" should be "prepare".
  2. section 2.3: please indicate the machine used for glow-discharging.
  3. section 2.4: please indicate dose rate (in e-/pix/s) at the camera level and exposure time for both dataset.
  4. section 2.4: please indicate beam settings for both data collection: micro- or nanoprobe, parallel illumination (or not), coma-free alignement procedure (or none).
  5. section 2.4: please indicate which software was used for drift correction.
  6. line 169: "resolution"; is that the resolution estimated by CTFFIND-4? Please specify.
  7. section 2.4.1: Was CTF refinement (magnification anisotropy, local defocus and astigmatism, coma) used for the apoferritin sample?
  8. section 2.4.2: The aberrations correction (magnification anisotropy, coma...) seemed to have a huge effect on the map quality, much more than with the apoferritin; could the authors comment on that? Was there a specific alignement/tuning issue on the microscope for this particular dataset that could be "rescued" later in silico?
  9. section 2.4.3: please indicate the software used for local resolution estimation.
  10. line 414: "et" should be "and".
  11. line 425: "outside" doesn't feel right, please rephrase.
  12. figure 5: a cross-sectional view of the local resolution would be useful with maybe a zoom-in on some of the metal cluster. Please also use a more conventional local resolution scale bar.

Author Response

Reviewer2

In this study, Cherrier et al. describe their setup for cryo-EM grids preparation in an anaerobic chamber ("glove box") for the purpose of studying the structure of oxygen-sensitive metalloproteins.

The paper reads well and introduces nicely and clearly for a non-expert audience the importance and specificity of the biochemistry of the metalloproteins and the methodological challenges encountered when studying them, particularly the necessity of working in a anaerobic environment.

The description of the setup for cryo-EM grid freezing in an anaerobic chamber (adapted from a cristallisation setup also developed by the lab) is clear and easy to follow for all people with minimal knowledge of the cryo-EM workflow. The important steps, specific bottlenecks and the ways to overcome them are well explained and described.

As a proof of concept, the authors also prepared and solved the structure of a well-known test sample aka apoferritin (the "lysozyme" of cryo-EM) to show the ability of the installation to produce "good" grids; and they also prepared and solved the structure of an oxygene-sensitive metalloprotein (PFOR) to prove that the anaerobic environment is working. Both cryo-EM structures were at rather high resolution (below 3 Å) and the procedures used and described (grid preparation, data collection, single particle analysis and model building) correspond to a "state-of-the-art" workflow.

Answer: We thank the reviewer for his/her kind comments

Major points:

  1. As a non-expert of the metalloproteins field, I think it is important to describe what kind of damage(s) could be expected from the oxygen action on a metal center; will it be destroyed? modified? What should we expect to see (or not see) in a (cryo-EM) structure done in aerobic conditions? What would be the minimal resolution to achieve to clearly see and assess such damages?

Answer: We modified the text accordingly, notably by adding references to papers describing such damages. Transition metals can be oxidized by molecular oxygen, leading to cascades of ROS production, with ensuing radical-based modifications at the protein. When considering [Fe4S4] clusters, one of the most abundant inorganic cofactors in nature, upon oxygen-induced oxidation from the +II to the +III states, one of the iron ions is rapidly expelled, leading to an unstable [Fe3S4]+ species with concomitant activity loss. This has extensively been characterized in aconitase but has been generalized to other [Fe4S4]-cluster containing proteins afterward. This unstable species is subsequently converted into a [Fe2S2] cluster with concomitant coordination geometry change from a cube to a rhomb, thus leading to extensive structural rearrangements. Different biological systems such as fumarate, nitrate reduction regulator (FNR) or IRP-1 (cytoplasmic aconitase) use these [Fe4S4] cluster properties to induce significant structural changes for regulation purposes. For instance, FNR is the molecular switch between anaerobic and aerobic metabolisms in facultative anaerobes. Under anaerobic conditions, FNR binds a [Fe4S4] cluster, is a dimer and binds DNA. Upon oxygen exposure, the cluster is destroyed, hence disrupting the dimer and impairing DNA binding. IRP-1 senses the iron content in the eukaryotic cells and, in the absence of cluster, the protein opens (broad domain rearrangements) and binds IRE (Iron Regulation Element) sequences on mRNAs to activate or repress protein expression. Therefore, while the slight chemical modifications cannot be accurately studied using cryo-EM, the broad structural changes induced by such cluster disruption can. Furthermore, when clusters are at the interface of domains or subunits, oxygen-induce damages will lead to disruption of such complexes or to the loss of subunit, thus leading to an increase in sample heterogeneity and of the particles on the grids. It seemed therefore important to us to develop a method to prepare cryo-EM grids under anaerobic conditions to preserve the metallocenter integrity and pave the way to a broader use of the technique to the structural characterization of proteins sensitive to oxygen.

  1. On a similar line; how well is PFOR suitable for such a study? It is mentioned that it is oxygen-sensitive but to what extend? I understand that a cristallisation experiment can last several days, even weeks, but freezing cryo-EM grids takes ~30 minutes: is this time frame still relevant in term of oxygen exposure, i.e. is that enough to "damage" the metal centres of the PFOR dimer?

Answer: We agree with the reviewer that many FeS-cluster containing proteins can be, and in fact have been, studied by cryo-EM without taking care of the FeS cluster integrity. The relative short time needed to freeze cryo-EM grids and the slight structural changes induced during that time may not be detrimental to such analyses. For example, over the last years, many EM structures of Complex-I or of the respiratory chain have been reported by different groups. As already mentioned in our answer to reviewer 1, in our experience, exposing crystals to oxygen for a few minutes before flash-freezing leads to a partial damage of the FeS clusters, hence inducing an increase in the B-factors and a blurring of the electron density of the surrounding residues. This should not be significant for most of the cryo-EM studies because the level of details cannot be accurately assessed due to moderate resolution. Yet, as mentioned above, when clusters are at the interface of domains or subunits, oxygen-induce damages will lead to disruption of such complexes or to the loss of subunit, thus leading to an increase in sample heterogeneity and of the particles on the grids. Even if PFOR is not the best suited sample when considering oxygen sensitivity, our proposed method led to high quality grids prepared in an anaerobic environment as already established by our long-standing experience in handling and crystallizing such oxygen-sensitive proteins, notably FNR, one of the most oxygen-sensitive protein we studied to date. Therefore, many different biological systems are still inaccessible to cryo-EM studies due to the lack of reliable methods to prepare grids in ana anaerobic environment. The most challenging part in our development was not to preserve anaerobic environment as it has been established that the use of dedicated gloveboxes is the solution. However, freezing grids in this anaerobic environment and preserving them from icing while taking them out of the glovebox was the trickiest part.

  1. In section "3.2.1 Apoferritin". I strongly disagree with the statement saying that the resolution is limited by the EM setup and that better resolution could be achieved by using a "state-of-the-art" electron microscope. One needs to compare what is comparable: the very-high resolution (sub-1.5 Å resolution) apoferritin structures cited by the authors are from a complete different source and definitely not horse spleen protein purchased from SIGMA. A quick research in the EMDB (filtering the horse apoferritin structures) shows on the opposite that the apoferritin structure reported in the present paper is on the high resolution side for such a sample, none of them going below 2.1-2.2 Å! This paragraph needs to be modified.

Answer: we slightly disagree with the reviewer on this point. On the one hand, the reached resolution when using horse spleen apoferritin purchased from SIGMA is comparable to that previously reported in the EMDB and in fact corresponds to the highest resolution structure we determined so far using our GLACIOS electron microscope, hence supporting that our anaerobic setup gives high-quality grids. On the other hand, we recently used the same apoferritin sample (different grid) at CM01 (ESRF, Grenoble France) on a KriosG3 with a K2 camera. The resulting 3D reconstruction reached 1.6A resolution (personal communication), illustrating that the microscope itself can be a limiting parameter.

Minor points:

  1. line 80: "prepared" should be "prepare".

Answer: Done

  1. section 2.3: please indicate the machine used for glow-discharging.

Answer: This has now been indicated (EMITECH)

  1. section 2.4: please indicate dose rate (in e-/pix/s) at the camera level and exposure time for both dataset.

Answer: Done (1 e--2.frame-1).

  1. section 2.4: please indicate beam settings for both data collection: micro- or nanoprobe, parallel illumination (or not), coma-free alignement procedure (or none).

Answer: We now added two sentences:

“Nanoprobe mode was used and the data collection was performed using parallel beam illumination.” & “SerialEM was also used to perform objective astigmatism correction as well as coma-free correction.”

  1. section 2.4: please indicate which software was used for drift correction.

Answer: This has now been indicated.

  1. line 169: "resolution"; is that the resolution estimated by CTFFIND-4? Please specify.

Answer: Done

  1. section 2.4.1: Was CTF refinement (magnification anisotropy, local defocus and astigmatism, coma) used for the apoferritin sample?

Answer: We now indicated that refinement of both the asymmetrical and symmetrical aberrations and the per-particle defocus values were performed

  1. section 2.4.2: The aberrations correction (magnification anisotropy, coma...) seemed to have a huge effect on the map quality, much more than with the apoferritin; could the authors comment on that? Was there a specific alignement/tuning issue on the microscope for this particular dataset that could be "rescued" later in silico?

Answer: There was no difference in the setting and the operator was the same. We do not understand why this happened.

  1. section 2.4.3: please indicate the software used for local resolution estimation.

Answer: Done

  1. line 414: "et" should be "and".

Answer: Done

  1. line 425: "outside" doesn't feel right, please rephrase.

Answer: This has been changed to “in the normal atmosphere”

  1. figure 5: a cross-sectional view of the local resolution would be useful with maybe a zoom-in on some of the metal cluster. Please also use a more conventional local resolution scale bar.

Answer: A figure with the angular distribution has been added in the supplementary material. A more conventional local resolution scale bar has also been added.

Reviewer 3 Report

Please see the attached Word file.

Author Response

Reviewer3

The manuscript by Mickael V Cherrier et al describes a procedure for preparing cryo EM grids of oxygen sensitive protein under anaerobic environment. Borrowing from their previous experience with sample preparation for cyrocrystallography, authors have customized a glove box to accommodate a Vitrobot for grid freezing, installed airlocks to transfer Dewar into the glove box and equipped it with necessary accessories for controlling nitrogen flow, humidity level or vacuum inside the box. Authors use this set up for preparing grids from commercially sourced apoferritin (as control protein) and from their 20 year old stock of PFOR protein (an oxygen sensitive protein). The cryo EM density maps obtained from these grids were modeled using previously published crystal structures.

Even though the manuscript demonstrates the utility of authors’ anaerobic set up for preparing cryo EM grids of above proteins and subsequently solving their cryo EM structures to high resolution, the manuscript is essentially a method paper offering no real innovation and no new structure. The kind of glove box tweaks described by authors to obtain anaerobic conditions are not unfamiliar to labs dealing with such proteins, neither have they solved an unknown structure with this approach. Although there is no doubt on the quality of data, the manuscript does appear to be re-inventing the wheel by re-solving two known structures by fitting known atomic coordinates to their cryo EM maps. The real demonstration of the utility of their approach would have been solving an unknown oxygen sensitive protein bottom up from cryo EM density maps to atomic coordinates.

Since the manuscript lacks new results and does not provide any real innovation, it is not suitable for publication in MDPI Biomolecules. This manuscript is suitable for publication in more specialized methods journals.

Answer: We disagree with the reviewer’s comments. We do not consider ourselves re-inventing the wheel because nobody so far reported any setup to successfully prepare cryo-EM grids under anaerobic conditions leading to high resolution cryo-EM structures. Our manuscript fulfills this purpose as illustrated by the use of two model proteins. Our method, as stated by the three other reviewers, “is of interest” and “may invoke further ideas in the future”. Hence, we consider our manuscript is interesting enough for MDPI Biomolecules readership. For instance, preserving the integrity of metallocenters is often key to understand the function of such metalloproteins even though resolutions usually accessible by cryo-EM may be lower than that reached when using X-ray crystallography. Nevertheless, when such metallocenters are at the interface between different subunits, oxygen induced damage may lead to rapid complex disruption or loss of subunits, hence leading to heterogenous samples. Sometimes, even if the structural changes cannot be detected by cryo EM, functional interpretation may be misled. We would cite the example of the SARS-COV-2 RNA-dependent RNA polymerase (RdRp). Two recent published cryo-EM structures reported zinc fingers (0.1126/science.abb7498 or 10.1038/s41586-020-2368-8). However, very recently, it has been reported that in RdRp active form, two [Fe4S4] clusters are present instead of zinc, leading to a therapeutic strategy based on the use of free radicals to destroy these clusters and impair viral development (10.1126/science.abi5224). Therefore, in order to determine the true active form of RdRp, one must work under strict anaerobic conditions and use a suitable protocol to prepare the corresponding cryo-EM grids. Unfortunately, because this setup is not easily accessible to the community, nobody performed such preparation so far. With the rise of cryo-EM as a major technique to determine protein structures over the last ten years, it seems important to us to propose such method to the community, hence paving the way to a better use of cryo-EM for the structural characterization of metalloproteins sensitive to oxygen-induced damages.

Reviewer 4 Report

The authors present a method for plunge-freezing cryo-EM samples in anaerobic conditions, which is important in the study of oxygen-sensitive enzymes. The method is adapted from their previous work on flash freezing of protein crystals.

The environmental setup is clearly presented, the experiments are controlled and the results are convincing. In particular, the appearance of all iron-sulfur clusters in the PFOR enzyme indicate that cryo-EM and anaerobic conditions were properly met. It is evident that the method described here works well for the authors. Additionally, the handling of the different cooling substances, the use of vacuum cycles, and plunging in a nitrogen environment with 1% relative humidity in the surrounding, are all interesting and somewhat uncommon in the single particle cryo-EM field. As described, they may invoke further ideas in the future. For these reasons the paper is worth publishing.

My main criticism is that the actual design described by the authors is of limited scope. It is limited to the use of the Vitrobot (or a custom-designed plunger with a similar cooling container) and to a short period of 30 minutes plunging. In that sense, it seems that improvements in the technique would be essential. The authors should make this clear upfront, discuss the weaknesses of the technique and, importantly, discuss alternatives. Unless the authors explain otherwise, there may be much simpler ways to plunge in an anaerobic environment with different plunging devices and for longer periods. For example, wouldn’t it be less cumbersome to run liquid nitrogen and ethane/propane gas pipes into the glovebox and liquefy the ethane there? The authors already connected the liquid nitrogen cylinder to the glovebox, so why not pump liquid nitrogen from there? Liquefying the ethane/propane inside the glovebox will allow longer working times, and the method can be adopted for using plungers of different designs such as the Leica EM-GP, Gatan Cryoplunge and homemade ones. Additionally, there will be no need to handle the Vitrobot cooling container outside the glovebox, which is a source of oxygen contamination. Pumping the LN2 directly to the glovebox may reduce oxygen contamination in the LN2 as described in the text (line 281). It is also possible to purchase a small 10L ethane cylinder, which can fit permanently in the glovebox. I noticed that in a previous paper that is cited in the current manuscript, the authors say that piping ethane and nitrogen “may lead to oxygen contamination” (Vernede and Fontecilla-Camps 1999), was this tested?

Other comments:

The paper lacks discussion and references to other methods used for environmental control of plunging. Such methods have been suggested almost since the invention of the plunger. For example, see this paper from Talmon and colleagues - Bellare et al. (1988) Controlled environment vitrification system: An improved sample preparation technique. Journal of Electron Microscopy Technique 10:87–111.

Please explain how the water for the Vitrobot humidifier is introduced into the glovebox? Are they treated in any special way?

The Vitrobot cooling container has two additional metal devices, the “Spider” which is used for cooling the ethane (and is removed to prevent freezing), and the grid box holder. Both are not mentioned in the manuscript and are not shown in the pictures. Are they used? It is also not clear how are the grids transferred to the cryo-boxes for storage following plunging (there are no cryo-boxes in the pictures), and how the cryo boxes are extracted from the glovebox.

Table 1 - defocus range missing a minus sign

Line 273 – change to “Absorbed”

Lines 240-302 – The advantage of using ethane over propane is that propane sublimes in vacuum at a lower rate compared to ethane. Therefore it may affect the amount of residual material left on the specimen. Do you see that in the microscope?

Lines 298-302 – If I understand correctly the term, the “white haze” is evaporated ethane, not water. If it were water then the whole cooling container would be quickly contaminated with ice crystals.

Line 408 - “The smaller size and lower symmetry of PFOR, when compared to apoferritin, explain the lower resolution”  - It is a totally different protein with different properties, not only smaller and less symmetric. Better to remove this sentence.

Figure 2 – please add a micrograph and FFTs to the figure to support the claim of a well-vitrified with nice particles sample.

Figures 3-5 – all images can be included in a single figure, as they all show steps in the analysis of the same structure. No need for 3 separate figures.

Please add an angular distribution plot (and make sure all angles are properly samples). Can be in supplementary.

Figure 3 – the isosurface view of PFOR looks noisy. Can a different threshold or different filtering be chosen to better display this model? Perhaps using local filtering. 

Figure 4 – the “corrected FSC” curve is missing. This is actually the most important one because it takes into account overfitting artifacts.  

Figure S4 – The EM density in C looks very unreliable. Maybe use some form of local reconstruction or local filtering to improve. Otherwise, I really doubt the reliability of the model there.

Most references to figures have 0 in them and I see no references to the supplementary figures.

Validation report - Clash score in the PFOR model is relatively high, please correct where possible.

Author Response

Reviewer4

The authors present a method for plunge-freezing cryo-EM samples in anaerobic conditions, which is important in the study of oxygen-sensitive enzymes. The method is adapted from their previous work on flash freezing of protein crystals.

The environmental setup is clearly presented, the experiments are controlled and the results are convincing. In particular, the appearance of all iron-sulfur clusters in the PFOR enzyme indicate that cryo-EM and anaerobic conditions were properly met. It is evident that the method described here works well for the authors. Additionally, the handling of the different cooling substances, the use of vacuum cycles, and plunging in a nitrogen environment with 1% relative humidity in the surrounding, are all interesting and somewhat uncommon in the single particle cryo-EM field. As described, they may invoke further ideas in the future. For these reasons the paper is worth publishing.

Answer: We thank the reviewer for his/her comments.

My main criticism is that the actual design described by the authors is of limited scope. It is limited to the use of the Vitrobot (or a custom-designed plunger with a similar cooling container) and to a short period of 30 minutes plunging. In that sense, it seems that improvements in the technique would be essential. The authors should make this clear upfront, discuss the weaknesses of the technique and, importantly, discuss alternatives. Unless the authors explain otherwise, there may be much simpler ways to plunge in an anaerobic environment with different plunging devices and for longer periods. For example, wouldn’t it be less cumbersome to run liquid nitrogen and ethane/propane gas pipes into the glovebox and liquefy the ethane there? The authors already connected the liquid nitrogen cylinder to the glovebox, so why not pump liquid nitrogen from there? Liquefying the ethane/propane inside the glovebox will allow longer working times, and the method can be adopted for using plungers of different designs such as the Leica EM-GP, Gatan Cryoplunge and homemade ones. Additionally, there will be no need to handle the Vitrobot cooling container outside the glovebox, which is a source of oxygen contamination. Pumping the LN2 directly to the glovebox may reduce oxygen contamination in the LN2 as described in the text (line 281). It is also possible to purchase a small 10L ethane cylinder, which can fit permanently in the glovebox. I noticed that in a previous paper that is cited in the current manuscript, the authors say that piping ethane and nitrogen “may lead to oxygen contamination” (Vernede and Fontecilla-Camps 1999), was this tested?

Answer: We disagree with the reviewer on these points. Our method to enter cryogenic fluids at desired temperature to prepare cryo-EM grids under anaerobic environment can actually be used with any plunger, provided slight adaptation to use the corresponding devices and tools. There is no limitation. In the present protocol, we can freeze grids during a 30-minute period before needing a cryogenic fluid refill. Such a need for refill also occurs in air. The only difference corresponds to the need to bring them in the glovebox. During this refill, the atmosphere of the glovebox in regenerated and the oxygen level drops back to 1-2 ppm. Therefore, a new cycle of grid preparation is possible, and so on all day long. This does not depend on the chosen plunger. The reviewer’s proposal to run liquid nitrogen and ethane/propane pipes inside the glovebox sounds in fact much more cumbersome and raises important safety issues. Notwithstanding the fact that it would require fresh liquid nitrogen as it is the main source of oxygen contamination, it would require a very specific design for the glovebox, hence losing the versatility and flexibility of our currently proposed method. We want to point out that we never said we connected the liquid nitrogen cylinder to the glovebox. Piping liquid nitrogen through the glovebox wall would induce massive evaporation upon cooling the line and thus an important increase in the pressure inside the anaerobic chamber risking a plexiglass panel break or a blast. When considering piping ethane/propane from outside, the same pressure issue may exist. In addition, it will consume more gas, regardless all the safety issues in case of leakage as these gases are explosives. Again, in our opinion, our method seems more versatile for a broad use by different groups provided they have a standard glovebox and any type of plunger.

Other comments:

The paper lacks discussion and references to other methods used for environmental control of plunging. Such methods have been suggested almost since the invention of the plunger. For example, see this paper from Talmon and colleagues - Bellare et al. (1988) Controlled environment vitrification system: An improved sample preparation technique. Journal of Electron Microscopy Technique 10:87–111.

Answer: We agree with the reviewer, but this is not the scope of our present paper. In fact, working in an environment with a very low humidity level may be advantageous considering the grid quality. But we did not test this and thus decided not to discuss this point.

Please explain how the water for the Vitrobot humidifier is introduced into the glovebox? Are they treated in any special way?

Answer: Water was extensively degassed through vacuum cycles and nitrogen bubbling to remove any trace of oxygen prior being poured into the Vitrobot reservoir. This is now described in the text.

The Vitrobot cooling container has two additional metal devices, the “Spider” which is used for cooling the ethane (and is removed to prevent freezing), and the grid box holder. Both are not mentioned in the manuscript and are not shown in the pictures. Are they used? It is also not clear how are the grids transferred to the cryo-boxes for storage following plunging (there are no cryo-boxes in the pictures), and how the cryo boxes are extracted from the glovebox.

Answer: We added two sentences: “The Vitrobot "Spider" tool is normally used to cool ethane. Here, it is used instead to heat and accelerate liquefaction of solid ethane by placing it a few seconds upside down on the brass cup” & “The water used to maintain humidity in the grid-blotting chamber was extensively degassed through vacuum cycles and nitrogen gas bubbling to remove any trace of oxygen prior to be poured into the Vitrobot reservoir. After being placed inside cryo-boxes, grids were transferred back, with the dewar, outside of the glovebox, through the airlock (without further vacuum cycle).”

Table 1 - defocus range missing a minus sign

Answer: Done

Line 273 – change to “Absorbed”

Answer: Adsorbed is the correct word. Adsorption is the sticking of atoms or molecules to a surface. The particles which get stuck on can be from a gas, liquid or a dissolved solid.

Lines 240-302 – The advantage of using ethane over propane is that propane sublimes in vacuum at a lower rate compared to ethane. Therefore it may affect the amount of residual material left on the specimen. Do you see that in the microscope?

Answer: Both propane and ethane will sublimate in the electron microscope and even if one is at lower rate compared to the other one, in the microscope after ½ hour the result will be the same.

Lines 298-302 – If I understand correctly the term, the “white haze” is evaporated ethane, not water. If it were water then the whole cooling container would be quickly contaminated with ice crystals.

Answer: We don’t exactly know but yes this is the most likely hypothesis.

Line 408 - “The smaller size and lower symmetry of PFOR, when compared to apoferritin, explain the lower resolution”  - It is a totally different protein with different properties, not only smaller and less symmetric. Better to remove this sentence.

Answer: We removed that sentence.

Figure 2 – please add a micrograph and FFTs to the figure to support the claim of a well-vitrified with nice particles sample.

Answer: This has now been added in supplemental figure S3.

Figures 3-5 – all images can be included in a single figure, as they all show steps in the analysis of the same structure. No need for 3 separate figures.

Answer: Figures 3 to 5 have been merged into a single figure (new Figure 4).

Please add an angular distribution plot (and make sure all angles are properly samples). Can be in supplementary.

Answer: The angular distribution has been added accordingly in supplementary figure S1).

Figure 3 – the isosurface view of PFOR looks noisy. Can a different threshold or different filtering be chosen to better display this model? Perhaps using local filtering.

Answer: The figure has been modified accordingly.

Figure 4 – the “corrected FSC” curve is missing. This is actually the most important one because it takes into account overfitting artifacts.

Answer: The corrected FSC curve has now been added to the plot (Figure 4c).

Figure S4 – The EM density in C looks very unreliable. Maybe use some form of local reconstruction or local filtering to improve. Otherwise, I really doubt the reliability of the model there.

Answer: The actual goal for this figure (now figure S7) was in fact to show that some regions in PFOR are not visible and notably the N-terminal part of domain VII. As a consequence, it was expected that in this region, likely more flexible a higher disorder would be observed. And this is the case for the helix on the right of figure S7c. However, even though many side chains were difficult to see, the main chain is clearly visible and unambiguous. As shown in figure 5a, most regions of the protein are well defined with main and side chains both visible. This is clearly stated in the figure caption.

Most references to figures have 0 in them and I see no references to the supplementary figures.

Answer: This most likely came from the use of different “word” versions. To avoid it to happen again in the present version, the links to figures and tables have now been replaced by text.

Validation report - Clash score in the PFOR model is relatively high, please correct where possible.

Answer: The structure was further refined accordingly to reduce the clash score, while not altering the other quality control parameters (Ramachandran plot, Rotamers outliers, MolProbity score…).

Round 2

Reviewer 1 Report

The authors have convincingly addressed my previous concerns. 

Reviewer 3 Report

Authors have, in their response to this reviewer's comments, built a strong case for suitability of this manuscript for publication in MDPI Biomolecules.